# From tiny to immense: Geological spotlight on the Alexander Mosaic (National Archaeological Museum of Naples, Italy) using non-invasive in situ analyses

Giuseppina Balassone[1,2,3]*, Piergiulio Cappelletti[1,2], Alberto De Bonis[1,2], Antonio De Simone[4], Diego Di Martire[1], Sossio Fabio Graziano[2,5], Celestino Grifa[2,6], Paolo Giulierini[7], Francesco Izzo[1,2], Alessio Langella[1,2], Mariano Mercurio[2,6], Vincenzo Morra[1], Mariateresa Operetto[7], Amanda Piezzo[7], Concetta Rispoli[1,2], Maria Verde[8]

1 Dipartimento di Scienze della Terra, dell'Ambiente e delle Risorse, University of Naples Federico II, Naples, Italy, 2 Inter-University Center for Research on Archaeometry and Conservation Science (CRACS), University of Naples Federico II—University of Sannio, Benevento, Italy, 3 National Institute of Geophysics and Volcanology (INGV), Vesuvius Observatory, Naples, Italy, 4 Department of Human Sciences, University Suor Orsola Benincasa, Naples, Italy, 5 Dipartimento di Farmacia, University of Naples Federico II, Naples, Italy, 6 Dipartimento di Scienze e Tecnologie, University of Sannio, Benevento, Italy, 7 Museo Archeologico Nazionale, (MANN), Ministry of Culture, Napoli, Italy, 8 Dipartimento di Architettura, University of Naples Federico II, Naples, Italy

* balasson@unina.it

⊙ OPEN ACCESS

**Data Availability Statement:** All relevant data are within the manuscript and its Supporting information files.

## Abstract

A key challenge in the art and archaeological field is the instrumental analysis of objects and materials while preserving their integrity. In this study, the world-renowned artwork Alexander Mosaic (The Issus Battle, collection of the National Archaeological Museum of Naples, IT), the most iconic representation of the face of the Macedonian king Alexander the Great coming from a Pompeii domus, was thoroughly analyzed with mobile and non-invasive methods, within a great project of restoration started in 2020. Representative areas of the Mosaic, overall consisting of ca. two million of tesserae, was studied by in situ videomicroscopy, infrared thermography (IRT), multispectral imaging, portable X-ray fluorescence (pXRF), Fourier transform infrared (FTIR) and Raman spectroscopy. Ten tesserae colors were discriminated, and hypotheses on their geological provenances are proposed. Plasters, mineral components and other substances of old protective materials were characterized. The information obtained with this approach paved the way to knowledgeable restoration.

## Introduction

The Alexander Mosaic (Fig 1A) is one of the most impressive artworks of the antiquity by any standard and the most important mosaic of the Roman age [1–9]. The image of Alexander depicted in the central scene of the mosaic is perhaps the most iconic and well-known representation of his face in ancient art.

**Funding:** National Archaeological Museum of Naples (MANN), project # 4913-P The funders had no role in study design, data collection and analysis, even though the decision to publish in joint paper was in agreement with the MANN's institution. Some Colleagues of the MANN staff, active during working at the mosaic, are also included as co-author of this manuscript (Drs. P. Giulierini, A. Piezzo, M. Operetto), and collaborated to a general supervision during the final preparation the paper.

**Competing interests:** The authors have declared that no competing interests exist.

The mosaic was discovered in the archaeological site of Pompeii (Italy) in 1831 and now is exposed at the National Archaeological Museum of Naples (*Museo Archeologico Nazionale di Napoli*, hereafter MANN). The MANN hosts wealthy collections coming from the archaeological excavations of Pompeii and Herculaneum sites, which destiny was signed by a thick layer of superheated pyroclastic material erupted from the Somma-Vesuvius volcano in 79 CE, that in the meanwhile preserved extraordinary evidence of ancient Romans' life [7]. The scene depicted in the mosaic was interpreted as the Battle of Issus (333 BCE, near the Turkish Syrian border), in which Alexander the Great and the Macedonian army defeated the Persian army led by Darius III, king of the Persians. In particular, the artistic detail of the dark tree with its bare branches contextualizes the fight episode at Issus; indeed, Arab and western texts dating back to the medieval tradition, and among them also the book of Marco Polo "*Il Milione*", recall the battle of Issus as the "battle of the dry tree" (or "the solitary tree"), introducing a precise reference to the only landscape element present in the mosaic [7]. Among the alternative

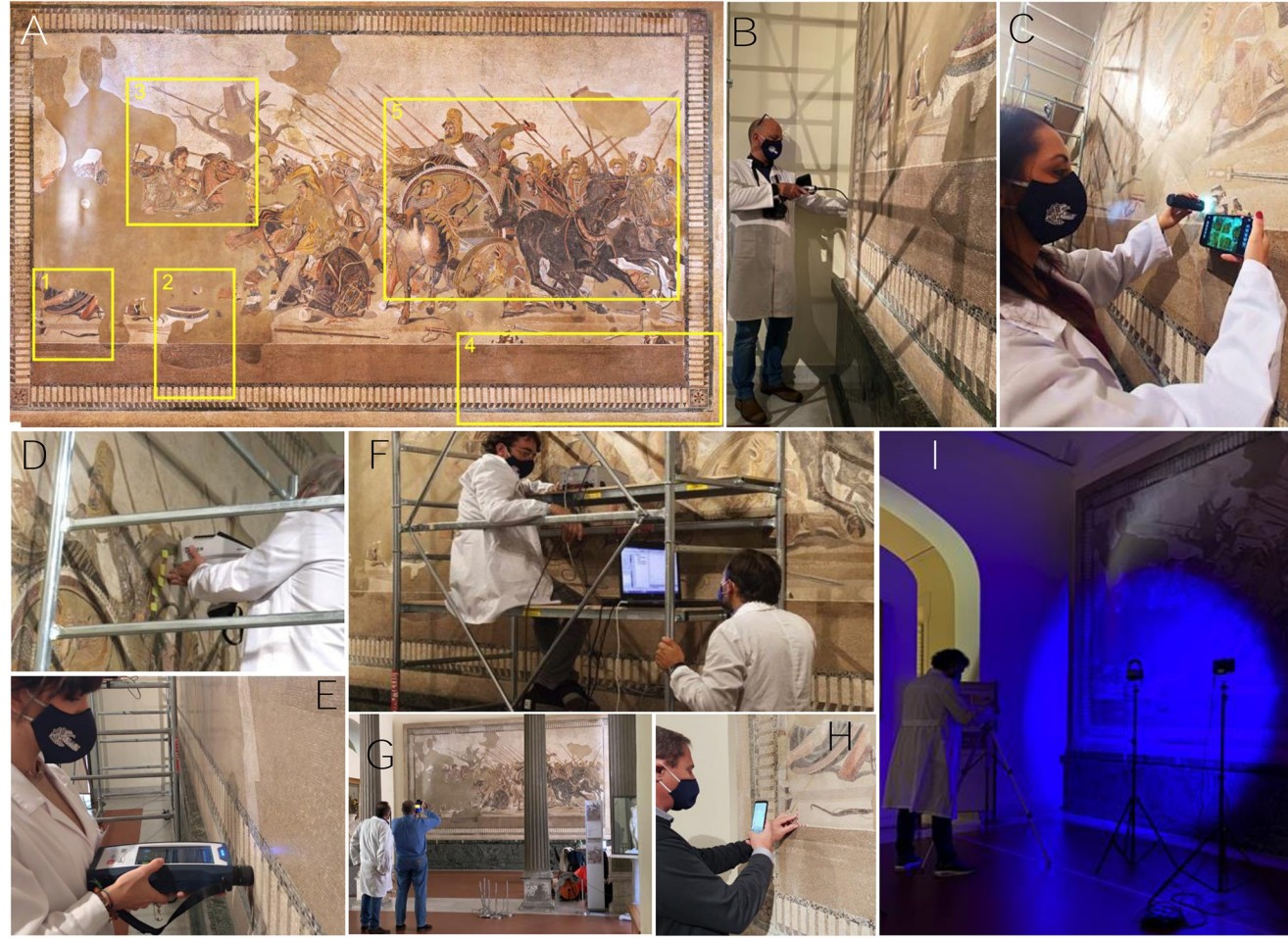

**Fig 1. The Alexander Mosaic (MANN, Naples, Italy) and in situ analyses.** The Alexander Mosaic and some pictures displaying the nondestructive noninvasive archaeometric campaign: A) The Alexander Mosaic (dimensions 583 x 325 cm) before the restoration intervention at MANN (September 2020), with indication of the examined areas (no. of point-analysis carried out by noninvasive techniques: area 1, 36 points; area 2, 23 points; area 3, 55 points; area 4, 106 points; area 5, 82 points). B) the endoscopic inspection of the back side. C) in-situ optical microscopy (OM). D) portable X-ray florescence (pXRF). E) Raman spectroscopy. F) External Reflectance Fourier transform infrared (ER-FTIR) spectroscopy. G) infrared thermography (IRT). H) tesserae counting. I) multispectral imaging.

hypotheses on the possible representation of the historical event there is also the battle of Gaugamela (331 BCE), in which, however, there is no reference to the dry tree [7].

The mosaic was found in the House of the Faun, one of the most luxury *domus* of Pompeii, as tessellated pavement located in the exedra facing the central peristyle. It is a reproduction, datable between the end of the 2nd century BCE and the beginning of the 1st century BCE, of a famous Hellenistic painting of 315 BCE most likely attributed to Filosseno of Eretria [7]. Millions of minute tesserae cover an area of exceptional dimensions measuring 583 x 325 cm and are arranged according to the extremely refined *Opus vermiculatum* technique which, using tiny tesserae (few millimeters), to obtain results very close to those of painting. The battle scene was carried out by artists of Hellenistic school with incomparable skill and care for details and has technical and stylistic similarities to the mosaic with a Nilotic subject that decorates the sanctuary of *Fortuna Primigenia* at Palestrina (near Rome, Italy). The Alexander Mosaic is a worldwide unicum, with an approximate estimation of the number of tesserae is ca. $1.9 \cdot 10^6$ of tesserae. Twenty-two years later from its discovery (1843), the former Bourbon authorities decided to remove the Mosaic along with part of the floor, screed from the Pompeian house and relocate it in the former Royal Archaeological Museum of Naples (today MANN) at that time, to preserve this outstanding artwork (Melillo, 2013). In this new location, the Mosaic was first horizontally set up on the ground floor of the Museum and then moved again in the current wall arrangement in the Museum "Mosaics Section" between 1916–1917. The state of conservation of the asset was profoundly affected by this new placement, with important instability phenomena [10, 11].

During the last decades, the MANN decided to carry out diagnostic investigations involving scientific institutions [12–15], aimed to assess the state of conservation of the Mosaic to plan possible restorations. Finally, during the 2020 the MANN launched the most important restoration project of the Alexander Mosaic. The restoration program provided a first diagnostic step carried out by the Center for Research on Archaeometry and Conservation Science (CRACS) which resulted in the application of noninvasive imaging and combined spectroscopic techniques. At the same time, a high-detail survey was realized by the University of Molise, obtaining a three-dimensional reconstruction of the internal mosaic structure [10]. In this work, we report the results of the noninvasive multi-analytical archaeometric investigations on the Alexander Mosaic, consisting of a combination of in-situ digital videomicroscopy and endoscopy, portable X-ray fluorescence, Infrared and Raman spectroscopy, multispectral imaging and infrared thermography (Fig 1B–1I). The so obtained data allowed to depict the chemical and mineralogical composition of the tesserae used for the manufacturing of the Mosaic, along with a tentative hypothesis on the provenance of the natural ones. In addition, an analytical evaluation of conservation state of the Mosaic was provided.

## Materials and methods

The entire surface of the Mosaic was preliminary explored using multispectral imaging techniques and, in this case, reflected and photo-induced luminescence images were captured [16]. In particular, the Mosaic was captured into 64 areas of about 1 m2 and for each of them we performed 1 visible reflected image (VIS), 3 infrared reflected images (IRR; 750-850-950 nm), 1 reflected ultraviolet image (UVR) and 1 induced luminescence image (UVL), for a total of 448 images.

Infrared thermography (IRT) aimed to detect surface thermal anomalies, mainly related to the conservation state of the Mosaic and was carried out on 6 selected areas (ca. 40 x 40 cm) by a high-sensitivity infrared thermal imaging FLIR T1030sc camera, with a $320 \times 240$ pixels infrared resolution and a temperature range between $-20$ and $650$ ˚C (with $\pm 2$ ˚C accuracy);

high-precision HDIR FLIR OSX™ optics are provided of ultrasonic focus system, drift compensation of external T and interference radiation protection. Thermograms were processed by the FLIR Tools software suite.

In-situ digital videomicroscopic observations (OM) of the Mosaic components (tesserae, mortars) were carried out via portable MIC-FI color digital microscope with Wi-Fi wireless transmission (smartphone/tablet-connected), white/UV/IR light and magnification 5x-200x, for a total of 94-point analyses.

Portable X-ray Fluorescence (pXRF) analyses for qualitative to semi-quantitative chemical compositions of tesserae (144-point analyses) were carried out by using a Bruker TRACER 5G spectrometer (Rh target X-ray source, silicon drift detector) operating at 30kV voltage, 10 μA current and 15 s acquisition time, with a 3 mm collimator under unfiltered conditions; pXRF spectra were processed with the software ARTAX Spectra 8.0.0.476 (Bruker AXS Handheld, Inc.) for the elemental identification and calculation of NET intensities of the identified elements, background and escape peaks corrections were applied to avoid misidentification of elemental peaks. Statistical analysis of the chemical data based on NET intensity data of 16 elements (Al, Ca, Cl, Cr, Cu, Fe, K, Mg, Mn, Ni, Pb, S, Si, Sr, Ti, Zn) was carried out via PCA (Principal Component Analysis) through processing with R Project software. Compositional groups were defined by confidence ellipses (95% confidence level for a multivariate t-distribution).

Fourier transform infrared (FTIR) spectroscopy was performed by a portable Bruker Optics ALPHA-R operating in External Reflectance (ER) mode (3 mm spot size); the instrument is equipped with the ROCKSOLID™ interferometer and the DTGS detector. Spectra were acquired in the wavelength range 7500–317 cm−1 at a nominal resolution of 4 cm−1 (1–3 min of acquisition time). ER-FTIR spectra were obtained on 44 different sites and then processed using the OPUS 7.0 software. Raman spectroscopy was carried out by means of a handheld Bruker Optics BRAVO spectrometer equipped with a double laser source (DuoLaser™) and 3 mm spot size, based on a patented technology (SSE™, Sequentially Shifted Excitation, patent number US8570507B1) for the reduction of the fluorescence effects; Raman spectra were processed using the OPUS 7.0 software. Analyses were performed on 20 points analyses.

Endoscopic investigations on the back sides, carried out all along the top, left and right sides of the Mosaic, were performed by a ZOTO Full HD 4-inch LCD digital endoscope, equipped with a digital inspection 1080P LCD camera, a 7 mm semi-rigid cable with waterproof borescope and 2x magnification.

## Results

Considering the dimensions of the Alexander Mosaic (ca. 19 m$^2$), a discriminating factor guiding the analytical protocol, and the investigated areas was strictly required. In the case study, the density of the spectroscopic analyses (Fig 1A and S1 Fig), aimed at characterizing the chemical-mineralogical composition of tesserae, was mainly driven by a preliminary visual inspection of the whole mosaic and more detailed multispectral imaging and infrared thermography. The latter techniques also provided an overview of the conservation state of the Mosaic and pivotal information about the treatments and deposits featuring its surface.

### Conservation state and surface materials

A fundamental preliminary aspect related to the diagnostics of the mosaic was the evaluation of its state of conservation and surficial treatments. This was achieved by investigating the entire surface down to more selected areas, also interpolating different instrumental analyses (Fig 2A–2K). In particular, among the multispectral imaging techniques, reflected and photo-

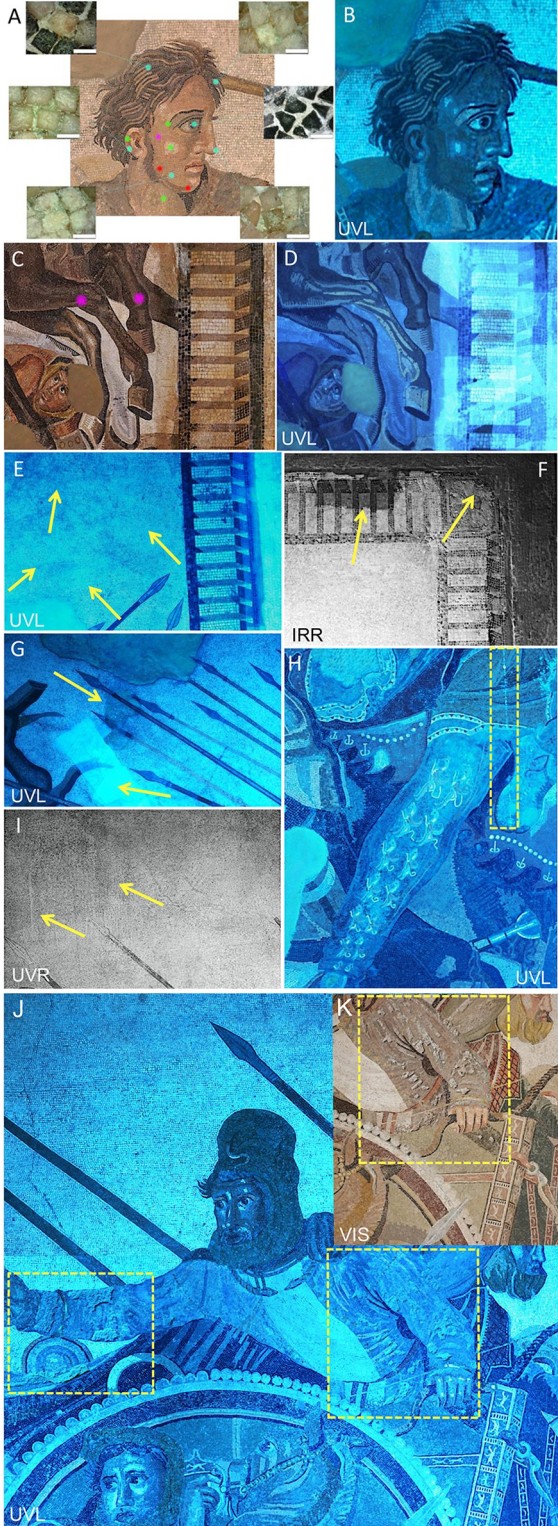

**Fig 2. Estimation of external treatments placement by multispectral imaging, with examples of interpolation of multi-technique study.** Estimation of external treatments placement by multispectral imaging, with examples of interpolation of multi-technique study. A) Alexander's face with point-analyses of micro-focus techniques (sky blue, OM; green Raman spectroscopy; magenta, FTIR; red, pXRF). B) the related induced luminescence effects (UVL) by multispectral imaging. C) The legs of a horse (magenta spots correspond to FTIR point-analyses), with D) the UVL

effects. E) UVL image of top-right area of the Mosaic, showing different luminescence effects (yellow arrows) due to compositionally heterogeneous binding mortars or to additional materials between the tesserae. F) IRR image of the tissue-paper facing (*velinatura*) of a frame sector (yellow arrows; see also Fig 5C and 5D). G) UVL image of *velinatura* restoration in the dry tree zone, together with a very discontinuous artificial adhesive patina (yellow arrows). H) Evidence of dripping of adhesive material (yellow dashed frame) used for consolidation in a UVL image. I) Heterogeneity of surficial treatments (yellow arrows) observed in a UVR image. J) UVL image of Darius III, showing a likely organic consolidant with a strong luminescence effect (yellow frames). K) Particular of the adhesive material on Darius's left arm, observed in VIS.

induced luminescence images captured on the entire surface of the mosaic provide, as a whole, information about the nature of the materials used for the manufacture of the Mosaic, as well as the occurrence of surface materials (both organic and inorganic) due to intentional treatments (*e.g.*, protective coatings, consolidants, pigments) and/or alteration processes (revealed by the presence of oxalates, sulphates, nitrates). Reflected images (VIS, IRR, UVR) are obtained when the wavelength of the incident radiation and the emitted radiation are the same. Among them, VIS images capture objects as seen by our eyes suggesting, for this case study, that tesserae were composed of ten types of colors masterfully combined to enhance artistic effects characterizing the Alexander Mosaic. In this case of photo-induced luminescence imaging, the wavelength of the emitted radiation is shorter than the wavelength of excitation (incident) radiation. Capturing images in the visible region by the luminescence (fluorescence) effects of a body irradiated with UV radiation (400 nm), it is possible to obtain UVL images to appreciate significant differences among the tesserae beyond what is visible.

The UVL images show that Alexander's face (Fig 2B), is composed of several shades of pink tesserae with appreciable changes in luminescence effects likely related to different chemical composition of the tesserae. A comparable effect on the luminescence can be observed on the *chiaroscuro* of the horse's legs at the right bottom of the mosaic (Fig 2C and 2D), also in this case likely due to compositional differences of lithotypes. The UVL images then guided the density of analyses, where the artistic effects were accompanied by tesserae with different luminescence response, including restored areas.

Multispectral imaging also highlighted the distribution of additional materials onto the surface of the Mosaic (Fig 2E–2K). Mortars with different compositions, as well as superimposed decorative layers, are evident by UVL measurements (Fig 2E), whereas the *velatino*, quite widespread as consolidation and restoration intervention, appears very clearly in UVL and IRR images (Fig 2F and 2G). Dripping and heterogeneity of the adhesive surficial materials can be noted in Fig 2H and 2I (UVL and UVR images), whereas a likely organic coating with a strong luminescence effect is observed in Fig 2I–2K (UVL, UVR and VIS images).

According to the combined use of the vibrational spectroscopic techniques (*i.e.*, Raman and ER-FTIR), the surface materials mainly consist of gypsum, natural waxes and traces of oxalates (S1 Table). The presence of gypsum is also confirmed by the very frequent occurrence of sulfur in pXRF measurements (S2 Table), along with the ubiquitous calcium signals. Gypsum was detected in OM as sporadic whitish deposits covering jointing mortars tesserae. Oxalates also occurs as the most stable calcium oxalate phase, namely whewellite [$Ca(C_2O_4)\cdot H_2O$]. As far as waxes are concerned, the typical spectral signal of natural ester-based waxes was identified characterized by the presence of a small derivative-like effect around 1740 cm$^{-1}$ in their ER-FTIR spectra (S2 Fig) attributable to the C = O bond vibration.

This feature allowed us to exclude the use of mineral wax (ceresin) or similar synthetic products, such as paraffin, and microcrystalline wax. However, it is hard to identify the exact composition of the natural wax applied [17] since this category of products (such as beeswax, carnauba, montana, candelilla wax, etc.) show signals that are quite comparable to each other,

or often compromised by spectral deformation and/or by the overlap with the other detected substances' absorption bands [17]. Actually, natural wax could not be ascribable to an original coating, as its low thermal stability [18] could not have withstood the high temperatures derived from the unexpected event of 79 CE.

IRT has revealed some thermal anomalies in different areas (Fig 3) for instance, around the black horse's head or near the dry tree trunk. It is worth noting that the IRT information pointed out that the areas affected by deterioration do not correspond to the ones that had been treated with the *velinatura*, which seems to be fulfilling its consolidation purpose. Rather, these anomalies could be attributed to volumetric changes in the mortars in response to temperature and humidity variations and/or deformations triggered by kinematic stresses (natural or artificial), inducing a depression in the right side of the mosaic, as also reported in [10]. This depression led to a non-perfect planar orientation of the tesserae, thus affecting the emissivity and consequently the recorded temperature.

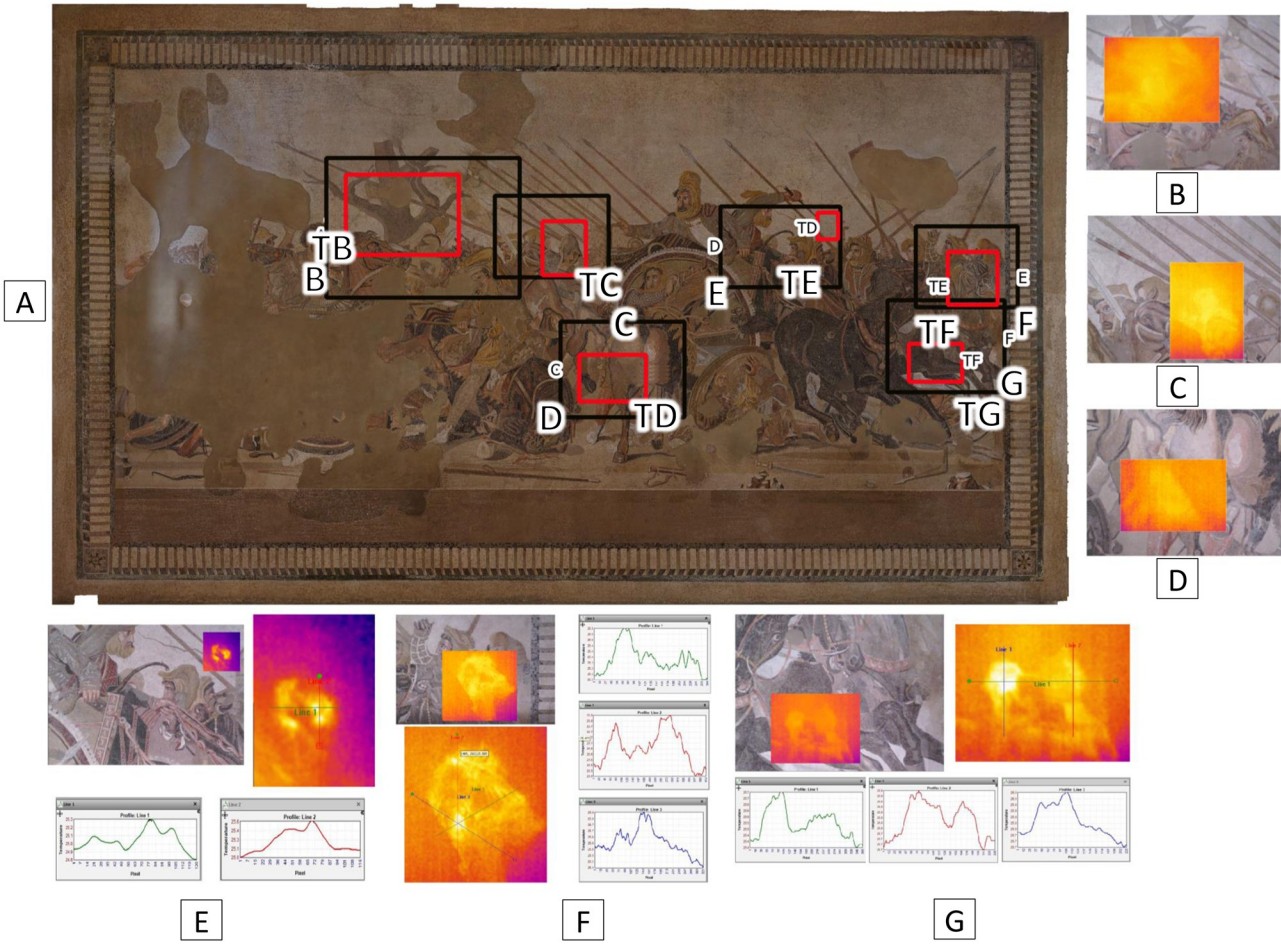

**Fig 3. Selected areas investigated by means of infrared thermography (IRT).** Selected areas investigated by means of infrared thermography (IRT). A) Portions of the mosaic highlight an anomalous behavior by means of infrared thermography (IRT). B) The dry tree area. C) The central warrior. D) The horse in close-up. E) The white tesserae in the Darius' charioteer area; the infrared image shows two linear plot lines, *i.e.*, a horizontal green line 1 and a vertical red line 2, respectively corresponding to the diagrams of temperature variation on the bottom side. F) The horse's head on the right side; the infrared image shows three linear plot lines, *i.e.*, the two oblique lines 1 (green) and 3 (blue) and the red profile vertical line 2, with the related diagrams of temperature variation shown at the right side. G) The black horse's forelegs at the right side; the infrared image shows three linear plot lines, *i.e.*, a horizontal line 1 (green) and two vertical red lines 2 (red) and 3 (blue), corresponding to the diagrams of temperature variation on the bottom side. In A), the labels B to G close to the black frames and TB to TG close to the red frames refer to the digital and IRT mages, respectively, shown in the related insets B) to G).

The endoscopic observations on the back sides (S3 Fig), among the wooden beams used for supporting the Mosaic in the current vertical position, as also detected by the georadar surveys [10] and inferred from the historical sources, detected many spaces filled with mostly paper, including newspaper; moreover, evidences of binder depositions are found, probably composed of vinyl or gypsum-based substances, likely in order to attempt to secure the structure of the Mosaic with a very fluid adhesive substance.

## Composition and textural features of tesserae

The results of the combined spectroscopic analyses are reported in this section with the aim of inferring the compositional features of the mosaic tesserae. As aforementioned, the Alexander Mosaic is composed of ca. $1.9 \cdot 10^6$ tesserae displaying ten kinds of colors and a wide range of micro-textures masterfully combined to enhance artistic effects of the artworks (Fig 4). These tesserae are joint by a whitish mortar well observed by means of OM images (Fig 4) showing faded chromatic variations resembling those of the tesserae (S4 Fig), likely made to give continuity to the depicted scene as in a painting, as also mentioned in literature [5]. Raman spectroscopy also detected traces of barium sulphates (*i.e.*, baryte $BaSO_4$) and lead carbonates (*i.e.*, cerussite $PbCO_3$) probably attributable, according to previous investigation [13], to contamination by restoration mortars.

Portable XRF measurements detected a very variable chemical composition of all tesserae in terms of type of elements and their NET intensities (S2 Table). Ca, Fe and S are ubiquitous (Table 1).

The occurrence of Ca is mostly ascribed to chemical composition of both carbonate-based components (*i.e.*, tesserae and/or possible contamination of jointing mortars) and, in the case of S, of sulphates as surface materials (together with oxalates). Fe is a common element both in

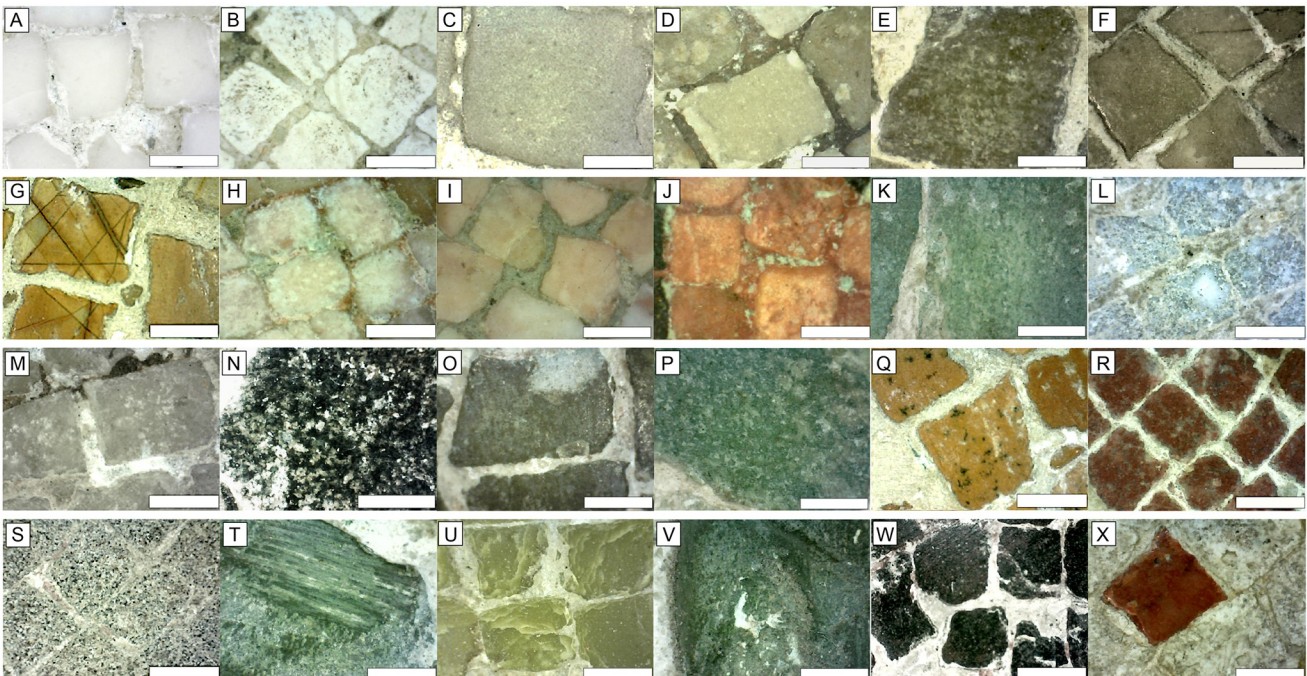

**Fig 4. Representative OM images of colored tesserae.** Representative OM images of colored tesserae belonging to Group 1 (A-M), Group 2 (N-P), Group 3 (Q-V) and Group 4 (W-X). Scale bar: 2 mm. Description of the groups in the text.

**Table 1. Relative abundances of the major chemical elements identified via pXRF in selected tesserae of different colors belonging to the four compositional groups.** References to OM images in Fig 4 are also reported along with the pXRF data points (see S2 Table for NET intensity data).

| Group | Colour | Image in Fig 4 | pXRF point | Chemistry by pXRF |
|---|---|---|---|---|
| 1 | White | A | XRF139 | Ca S Fe Cl Ti Sr Ni K Si Pb Zn Mn |
| 1 | White | B | XRF65 | Ca S Fe Cl Ni K Sr Pb Si Mn |
| 1 | Dark white | C | XRF14 | Ca S Cl Fe Ni K Sr |
| 1 | Dark white | D | XRF109 | Ca Cl Fe Ni S Sr |
| 1 | Brown | E | XRF7 | Ca S Fe Cl Ni K Sr Si |
| 1 | Brown | F | XRF108 | Ca Fe S Cl Ni K Sr Si Pb |
| 1 | Yellow | G | XRF38 | Ca S Fe Cl Ni Sr K Si Pb |
| 1 | Pink | H | XRF40 | Ca S Cl Fe Ni Sr Si Pb |
| 1 | Pink | I | XRF96 | Ca Fe S Mn Cl Ni Sr Si K Pb |
| 1 | Red | J | XRF90 | Ca Fe S Cl Ni Pb K Sr |
| 1 | Green | K | XRF57 | Ca Fe S Mn Cl Ti Ni Sr Pb K Zn |
| 1 | Blue | L | XRF73 | Ca S Fe Cl K Sr Ni Si Pb Ti Mn Cu |
| 1 | Grey | M | XRF39 | Ca S Fe Cl Ni Sr Ti K Pb |
| 2 | Black | N | XRF26 | **Fe Ca K S Sr Si Ti Cl Mn Pb Zn** |
| 2 | Grey | O | XRF33 | Fe Ca S K Sr Si Ti Cl Mn Ni Pb |
| 2 | Green | P | XRF127 | Fe Ca S K Si Sr Cl Ti Mn Pb Ni |
| 3 | Yellow | Q | XRF137 | Ca Fe S Pb K Ti Zn Si Cl Sr Ni Mn Cr |
| 3 | Red | R | XRF136 | Ca Si Fe S Mn Cl Ti Ni Sr Zn Pb |
| 3 | Grey | S | XRF51 | Ca S Cl Fe Sr Ni K Pb |
| 3 | Green | T | XRF123 | Ca Fe K Si S Cl Sr Ti Zn Cr Pb Ni |
| 3 | Green | U | XRF87 | Ca Pb Cu Fe S Si Cl Sr Mn Ni Ti |
| 3 | Green | V | XRF130 | Ca Pb Cl S K Sr Fe Ni Ti |
| 4 | Blue | - | XRF58 | Ca Fe S Si Cl Pb Mn K Sr |
| 4 | Blue | - | XRF59 | Ca Fe S Pb Si Cl Mn K Sr |
| 4 | Blue | - | XRF141 | Ca Fe S Si Cl Pb Mn K Sr |
| 4 | Black | W | XRF66 | **Fe Ca Mn S Si Cl Pb Sr Ti Ni** |
| 4 | Red | X | XRF132 | Fe Ca Mn S Cl Pb Si |

carbonate and silicate rocks. According to the PCA statistical treatment of pXRF dataset (Fig 5), Ca and Fe are the elements that most affect the dataset influencing the contribution of other elements on the total variance. First and second components (PC1 and PC2) have a very good score, responding to the 96% of the total variance, 80.8% and 15.1% respectively (Fig 5). The statistical treatment of pXRF data permitted discrimination in four groups of tesserae with the only exception of an outlier represented by a single green tessera.

**Group 1. Carbonate-bearing tesserae.** This group gathers most of the analyzed tesserae, which are correlated to the presence of calcium as suggested by the PCA (Fig 5). Although Ca is ubiquitous in all measurements due to sulphates/oxalates phases and in some instances jointing mortars spread onto the surface of the Mosaic, from the chemical point of view Group 1 (Fig 5) differs for a very high intensity of Ca (Ca Kα NET intensity > 150000; S2 Table). Raman and ER-FTIR measurements confirmed the carbonate composition of the natural tesserae belonging to this group (S1 Table and S2 Fig). Digital OM highlights that Group 1 includes all the ten types of colors observed during the preliminary visual inspection. NET intensities of some secondary characterizing elements seem to increase according to the type of color.

The Ca-rich composition is evident for white-colored tesserae (Fig 4A and 4B) due to the prevalence of calcite; no significant chemical variations were observed for other elements

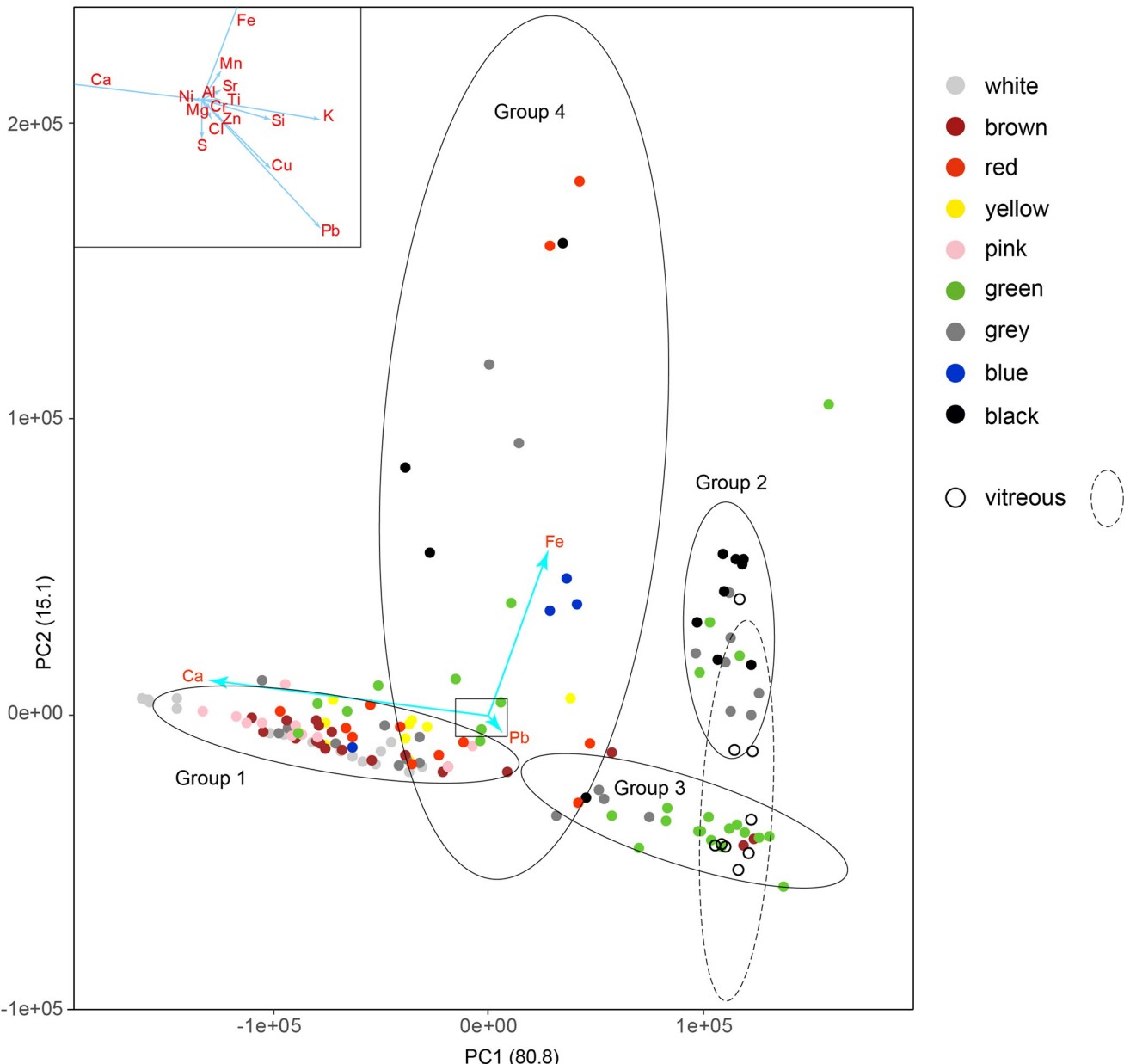

**Fig 5. Compositional features of the Mosaic tesserae of different colors pointed out by PCA on chemical data.** Compositional features of the Mosaic tesserae of different colors pointed out by PCA on chemical data. Confidence ellipses define four compositional groups of tesserae: carbonate-bearing (Group 1), silicate-bearing (Group 2), intermediate composition and vitreous (Group 3), iron-bearing (Group 4). Data of vitreous tesserae from other mosaics exposed at MANN were also reported along with the related confidence ellipse defining the group.

(Table 1). Two different types of white-colored tesserae can be recognized: a first type, analyzed in the shield at the bottom of Alexander figure, is distinguished by the polished and homogeneous fine-grained texture (Fig 4A), a second one was analyzed in the Darius'armor, and shows a rougher surface with scattered tiny blackish inclusions (Fig 4B).

In dark white tesserae (Fig 4C and 4D) slightly higher Mg intensities compared to those found in Ca-rich tesserae were observed (S2 Table), suggesting the presence of Mg-rich

carbonate. Measurements of the dark white tesserae were performed on the frame (Fig 4C) and in the basal band (Fig 4D) of the mosaic, both showing a rough surface with a microcrystalline texture and similar features of a fine-grained (micritic) limestone. A very similar composition also characterizes the brown tesserae localized in the frame (Fig 4E) and in the basal band (Fig 4F). They show a micritic texture (Fig 4E and 4F) and in some cases exhibit a polished surface (Fig 4F).

Yellow-colored tesserae feature the golden decoration on the soldier's helmet behind Alexander. They exhibit a composition in line with the previous ones, apart from a higher intensity of iron (S2 Table). They show a very characteristic texture made up of a regular pattern of tiny black stripes (Fig 4G).

Characteristic XRF lines of iron are more pronounced in red and pink tesserae, along with moderate signals of Mn in some pink-colored ones (S2 Table). Two types of pink tesserae were employed to represent the skin color of the figures, the first type (Fig 4H) is a fine-grained light pink rock localized on the Alexander's cheek, the second type (Fig 4I) is a microcrystalline rock with a more intense pink hue analyzed on the armor of the soldier to the right of Darius.

Red tesserae analyzed on the armor of the coachman to Dario's right, show a texture ranging from fine-grained (with black inclusions) to microcrystalline (Fig 4J).

Included in Group 1 there are also examples of green carbonate-bearing tesserae characterized by major concentrations of Ca and Fe and subordinate Mn, Ti, and Ni (S2 Table). They represent a detail of the shield located at the bottom left of the mosaic and show a micritic texture (Fig 4K).

Carbonate-bearing tesserae also include light blue tesserae (Fig 4L) included in the Darius' chariot, black tesserae (for instance, around Darius), and grey tesserae (Fig 4M) analyzed on the horse' face behind Alexander. Light blue tesserae are distinguished by the predominant presence of Ca and subordinate Fe and K. All of them have a polished appearance and a microcrystalline texture (Fig 4L and 4M).

**Group 2. Silicate-bearing tesserae.** This group is formed by tesserae with a silicate-bearing composition displaying black, grey, and green colors (Fig 5). The black tesserae (Fig 4N) show a microcrystalline texture with variable content of black to whitish inclusions. Most of these tesserae show high intensity of Si (S2 Table), along with Fe, Ca, K, and Ti, likely due to silicates such as pyroxene and feldspar, as suggested by vibrational spectroscopy (S1 Table). The tesserae are characterized by different sizes, the larger ones are part of the outer frame of the mosaic (Fig 4N), the smallest ones were used to highlight the characters' outlines and the most distinctive details (*e.g.*, lips, eyebrows and eyes) with the purpose of creating a stronger contrast of the figures.

A very similar composition also characterizes the grey tesserae localized in the frame. They are characterized by rocks with fine-grained (Fig 4O) to aphyric/glassy-like textures.

The green tesserae (Fig 4P) included in the floral decoration in the frame show aphanitic structure and high Si and Fe levels, which might point to a possible greenish silicate-rich rocks.

**Group 3. Natural and vitreous tesserae with intermediate composition.** Tesserae having an intermediate composition between the carbonate and silicate tesserae are included in Group 3 (Fig 5), which gathers tesserae of different colors: black, brown, yellow (Fig 4Q), red (Fig 4R), grey (Fig 4S), and green (Fig 4T–4V). In addition to the ubiquitous Ca and Fe, these tesserae generally show more pronounced amounts of Si and Mn. Relatively high Pb was detected in the yellow tesserae localized in the shield at the bottom of Alexander figure, which show black spot on the surface (Fig 4Q). The red color analyzed in shield at the bottom of the Alexander figure is represented by tesserae with high amounts of Ca, Si, and Fe and fine-grained texture (Fig 4R). The grey tesserae analyzed on the tip of the Persian spear apart from Ca, differ for higher intensities of S, Cl, Fe (Table 1) and a fine-grained texture (Fig 4S).

Other tesserae of the group 3 can have a vitreous appearance; among these, several green tesserae can be seen in the investigated areas (*e.g.*, Fig 4T–4V), although this color cannot be exclusively associated with the vitreous type but also with fragments of likely natural origin (see the section "Geological materials and sourcing of tesserae" in the Discussion paragraph). Green tesserae were analyzed on the frame (*e.g.*, Fig 4T), on the Persian soldier's arm (*e.g.*, Fig 4U), and in the floral decoration in the corner of the frame (Fig 4V). Tesserae like those of Fig 4T show Ca, Fe, K, and Si as main elements, which could be associated to natural rocks (*e.g.*, serpentinite). Other tesserae display a peculiar chemical signature, characterized by the presence of Pb, Ca, Cu, and in several cases Cl (Table 1); other minor elements (Fe, K, Si) were also detected. They plot on the right side of cluster 3 and are overlapped, at least in part, by data of artificial vitreous tesserae from other mosaics exposed at MANN, that were analyzed for comparison (Fig 5).

Two brown tesserae (points XRF110 and 111; S2 Table) ascribed to restoration works also plot in this area (Fig 5), pointing to their synthetic origin.

**Group 4. Iron-rich tesserae.**   Tesserae featured by a variable amount of iron belong to the group 4 (Fig 5). They are represented by light blue, grey, black, green, and red tesserae that scatter in the biplot. Light blue tesserae are clustered and, from the chemical point of view, they mainly show Ca with subordinate Fe and Si; Pb and Mn are also present, whereas ER-FTIR analysis shows ubiquitous calcite peaks.

Tesserae plotting in the upper sector (Fig 5) are featured by the highest amount of Fe and Mn and include black tesserae (Fig 4W) analyzed on Darius' chariot and the red one (Fig 4X) placed in the center of the floral decoration in the lower right-hand corner of the mosaic. They both show an fine-grained textures.

## Discussion

### The nature of the coating materials

Analyses of the materials found onto the surface of the Mosaic display an almost ubiquitous presence of calcium sulphate (gypsum), natural wax as well as traces of calcium oxalate.

The use of wax as a protection—for modern and ancient mosaics and paintings—is well-documented and recognized in archaeometry and conservation science. Wax is a natural product, of animal or plant origin, widely used as a protective coating for its compatibility with all kinds of materials (natural stone, wood, cellulosic materials, etc.). Spectroscopic studies [19] suggest that waxes can be easily distinguished from other organic substances (*e.g.*, proteins and resins), whenever used as protective or strengthening agents in cultural heritage [20, 21]. The wax coating detected from this study, could have been applied on the Mosaic as a part of the restoration works occurred over time, both before and after the detachment from the archaeological site and its transfer to its current position at the MANN. To support this hypothesis, Melillo in 2013 [5] reports some steps of the restoration work carried out until 1843 by the Neapolitan mosaicist Piedimonte. He used to conclude the cleaning process by applying a wax diluted in *aqua regia* onto the surface of the Mosaic after filling the lacunae with a slurry made of gypsum. According to a note of the architect Pietro Bianchi in 1827, this treatment, along with the high humidity of the site, led to the progressive deterioration of the Alexander Mosaic [5]. Then, the famous Roman mosaicist Vincenzo Raffaelli, director of the Vatican mosaics in Rome, was chosen by the Minister of Home Affairs to examine the conservation state of the Alexander Mosaic. Raffaelli found a coating that was similar to a "resinous varnish", responsible for the decay of the colors of the tesserae and of the general bad state of the artwork; he also pointed out the use of three different types of bedding mortars for the repositioning of the mosaic tesserae, *i.e.*, a lime-based mortar, that was applied originally (probably compatible

with the *supranucleus* that was typically produced for the bedding of Roman mosaic tesserae [22], a greyish lime mortar, used for more particular and detailed repairs, and a third type of mortar, made of a gypsum-based grout. According to Raffaelli, the use of gypsum was not appropriate in humid places because it could cause expansions and detachments around the gripping surface [5].

The widespread occurrence of gypsum detected on most of the areas analyzed using pXRF, Raman and ER-FTIR can be attributed to remnants of the gypsum applied, along with layers of paper and fabric, to protect the surface of the Alexander Mosaic during the phases of detachment and displacement of the artwork in 1843. In fact, when the mosaic was placed in a wooden box and moved to the *Real Museo Borbonico* in November 1843, during transportation it stayed in direct contact with the gypsum layer inside the box for several days and was also affected by heavy rain. Such prolonged contact between the Mosaic and the gypsum could have led to the additional precipitation of gypsum on the Mosaic and its wax coating, permeating the surface through the outermost part of the *supranucleus*. This may have caused the formation of a thin layer of gypsum, which is often detected by ER-FTIR spectroscopy as a reflective optical surface, in which the specular components distinctly prevail over the diffusive ones, unlike what one might expect from a gypsum normally scattered in a rough material such as mortar, when it occurs as a binder or a product of alteration [23, 24].

Oxalates could be the result of both the alteration processes of the organic protective coatings [18].

## Geological materials and sourcing of tesserae

According to Marcaida et al. [25], Pompeian mosaics—the most important decorative patterns in Ancient Roman cities, then in the Bay of Naples and surroundings of Mount Vesuvius—have not been deeply analyzed from a chemical point of view and only few works report the chemical and mineralogical composition of some tesserae from Pompeian mosaics [18, 26, 27]. The materials used for the realization of the Roman tesserae, made up of individual pieces of cubic form, are mainly composed by carbonate and volcanic rocks, colored glasses or ceramic materials, whereas the base materials used to join tesserae were lime and/or other binders [23, 25, 28, 29]. The tesserae used in Ancient Rome were mainly made of calcareous rocks obtained from local sources of natural stone, with additions of crushed brick, tile and pottery creating colored shades of dominant black, red, white, blue and yellow, commonly forming polychrome patterns [25].

The identification of base materials of the tesserae and their supply sources are a challenge for the archaeometric research, especially when it is not possible to take advantage of laboratory (more or less) destructive quantitative analyses (*i.e.*, thin section, X-ray powder diffraction, isotopic analyses) on (micro)samples. From a mineralogical-petrographic perspective, our data show that the investigated tesserae of the Alexander Mosaic belong to four compositional types: carbonate-bearing tesserae (Group 1), silicate-bearing tesserae (Group 2), natural and vitreous tesserae with intermediate composition (Group 3) and iron-rich tesserae with relatively high Fe concentrations (Group 4). Vitreous (*i.e.*, synthetic) tesserae are characterized by a peculiar composition as shown in Fig 5, and mostly include grey ad green colors, with the green ones richer in Cu (S2 Table). For comparison, portable XRF measurements were performed on blue and green vitreous tesserae from other mosaics exposed at MANN (Mosaic Room 59). The analyses, reported in Fig 5 with black circles mostly included in the dashed ellipse, confirm this assumption.

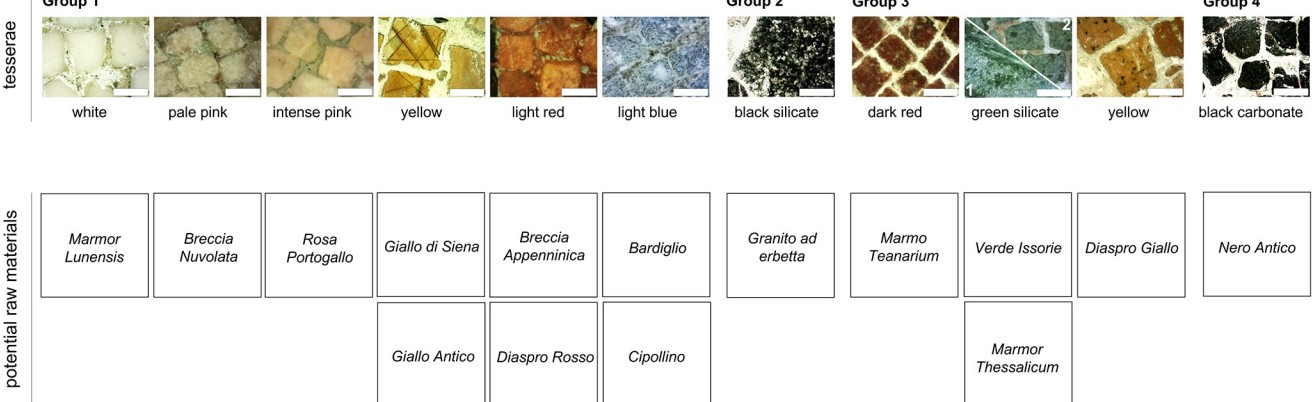

**Fig 6. Some hypothetical geological provenances of geomaterials.** Some hypothetical geological provenances of geomaterials used for the tesserae, based on qualitative remarks and similarity with more likely lithotypes used in Roman times [31, 32] (see text). For the two green tesserae of Group 3 (# 1 and 2) see text explanation. The white scale bars on the tesserae images are 2 mm.

Concerning the geological provenance of geomaterials, we can only speculate about possible source areas for some of the previously mentioned groups, considering qualitative remarks and similarity with some lithotypes likely employed at Roman times [30–35] (Fig 6).

Group 1 includes tesserae of various colors. Among these, the following can be ascribed to famous lithotypes.

Some of the white tesserae (Figs 4A and 6) could be related to the *Marmor Lunensis* from the Apuan Alps quarries (Italy), which started to be mined for marble in the 1st century BCE and was no longer used starting from the 3rd century CE; currently the analogous that is extracted in the Carrara basin is Bianco Carrara, in the C and CD varieties which can be distinguished by the presence of a whiter (C) or slightly grayer (CD) background color.

Pale pink tesserae (Figs 4H and 6) can reasonably be associated with the *Breccia Nuvolata* lithotype is quite reasonable, despite historical sources reported it was extracted starting from the 2$^{nd}$ century CE from various sites of the Mediterranean area (Italy, Tunisia, Algeria, and Libya); this material was used in several different ways in *Leptis Magna* (Libya) and even some walls of the Temple of Serapis in Pozzuoli (near Naples) are covered with this rock.

Intense pink tesserae (Figs 4I and 6) could be ascribed to the *Marmo Rosa* of Portugal, but a further origin could be the *Breccia Appenninica* (Apuan Alps); the absence of white veins, typical of the so-called *Portasanta* lithotype (*Marmor Chium*), does not support this hypothesis, although this lack could depend on the small size of the tesserae.

Some of the yellow tesserae (Figs 4G and 6) show blackish lineation as in the *Giallo di Siena* stone [31–33]. Other possible raw material could be the so-called *Giallo Antico* (*Marmor Numidicum*), extracted starting from the 2nd century BCE until the 3rd century CE in the roman city of *Simitthus* (modern *Chemtou*, Tunisia).

Light Red tesserae (Figs 4J and 6) could belong to the aforementioned *Breccia Appenninica* from Apuan Alps, or to a variety of *Diaspri Rossi* (Sicily).

Light blue tesserae (Figs 4L and 6) are composed of a carbonate lithotype with minor inclusions containing Si and Fe; this feature could be related to a marble like *Marmo Bardiglio* or *Cipollino*.

Group 2 mostly includes black and grey tesserae and some green ones. Black silicate tesserae (Figs 4N and 6) can be reasonably ascribed to a melanocratic microcrystalline rock such

as the so called *"Granito ad erbetta"*, a very fine-grained magmatic rock of gabbroid composition [36].

Group 3 gathers tesserae of different color including black, brown, yellow, red, grey, and mostly green.

Dark red *tesserae* (Figs 4R and 6) could possibly be attributed to the *Rosso Antico* (*Marmor Taenarium*, originating from Cape Matapan, Greece), known from the Romans since the end of the 2nd century BCE. This marble derives from low-grade metamorphism on impure carbonates: in addition to calcium carbonate, it also contains silica. The predominant mineralogical constituent of the lithotype is calcite with subordinate constituents as quartz, plagioclase, mica, chlorite [37].

For the green types, Fig 6 reports the green silicate tessera of Fig 4T (see also Table 1) and indicated with #1, as well as another green proxy indicated with #2. These lithotypes could have as a possible origin from the so-called *Verde Issorie* (a serpentinite rock). Also, the *Verde Antico* (*Marmor Thessalicum* or *Marmor Atracium*, originating from different areas of the Thessalia region) cannot be ruled out, being this rock mentioned in Diocletian's edict; it is a compact and fine-grained marble, with a bright green matrix and showing different hues, with the darker shade variety being quite rare.

Yellow tesserae from this group (Figs 4Q and 6) could be related to the *Diaspro Giallo/ Giallo Venato*, supplied from Spain or Sicily (Italy), which also shows the previously mentioned black spots. It should be remarked that variations in the hues of the yellow tesserae, but in general for all the color here discussed, could be obtained depending on different surface polishing degree.

Group 4 includes tesserae with a relatively high amounts of iron and different colors consisting in light blue, grey, black, green, and red.

Black carbonate tesserae of this group (Figs 4W and 6) are likely represented by fine-grained and compact carbonate types, rich in Fe and carbon-bearing inclusions, which could be assigned to the *Nero Antico* (Djebel Aziz, Tunisia). The use of an analogous black lithology from the Tunisian Chemtou site, where *Giallo Antico* was also extracted, cannot be excluded. This particular typology of tessera, with size less than 2mm, denotes, even more than the others, a strong compositional interaction with the (red-coloured) mortar surrounding it. This evidence can be seen on the Fig 4W and is reflected in the particularly high amount of Fe in the XRF analysis.

## Conclusions

In this study, data obtained with the in situ non-invasive and non-destructive analytical strategy provided both qualitative and (semi)quantitative archaeometric information on the various components of the Alexander Mosaic at the micro- to macroscopic scale. This information is primarily aimed at the assessment of its conservation state and the nature of the tesserae for the restoration work currently underway.

The key points can be summarized as follow:

### The Mosaic's surface

Combined in-situ analyses (OM, ER-FTIR, Raman, pXRF, multispectral imaging, IRT) revealed many details of selected surficial areas of the mosaic. The investigations performed through multispectral imaging have shown high luminescence both in limited areas, due to the compositional varieties of the mosaic tesserae, and in irregularly large areas, due to protective coatings. The *velatino* was a quite common consolidant for the restoration intervention. Infrared thermography revealed some areas with temperature variations, but these did not

correspond to areas treated with *velinatura* consolidation, which appears to have fulfilled its purpose. However, in assessing the conservation state, thermal imaging revealed some critical areas in different parts of the mosaic, which deserve special attention.

Gypsum, natural wax, and traces of calcium oxalate are the common phases detected on the Mosaic's surface. It is likely that wax, a common protective coating for artworks, was applied during the restoration process, as the restorers used to do in the past. Calcium oxalate, found in traces, may be a by-product of the degradation of organic protective coatings. Gypsum, which is widespread throughout the Mosaic, probably originated from protective layers applied during detachment and transport at the Museum in the 19th century, resulting in a thin layer of this component in many areas of the surface.

## The Mosaic's tesserae and mortars

A careful micro and macro inspection of the tesserae along with spectroscopic analyses allowed for drawing some hypothesis on their geological nature and tentatively assign potential provenance areas. Further than surficial investigation endoscopic observations from the back side of the artwork detected many empty portions, probably representing areas that were not reached by gypsum-based substances added to secure the structure of the mosaic during the former dismantling from its original site and transport operations. These potential areas of weakness should be taken into due consideration during restoration.

Based on OM, pXRF with PCA data processing, ER-FTIR and Raman analyses, the investigated tesserae can be sorted into four main groups: calcium carbonate-based (the most heterogeneous group and with many colors, *i.e.*, black, white, grey, red, brown, pink, yellow, green, light blue), silicate-based (grey, black, green), a mix of both and vitreous types (black, grey, brown, yellow, red, green), and Fe(Ca+Mn)-bearing tesserae (black and red, but also grey, green, light blue). Microcrystalline texture is more widespread among the considered lithologies. As far as the possible origin of the rock raw materials that made up the mosaic tesserae is concerned—a challenging topic in archaeological provenance studies—only hypotheses can be made, given the completely non-destructive mode of the study. Based on lithological similarities with rocks from known mining districts of Roman times, several mining areas in the Mediterranean region can be considered as geological sources, such as Italy, Greece, Iberian Peninsula and Tunisia. The jointing mortars can have different compositions (even though commonly Ca-bearing) and colors, which vary from whitish to different shades sometimes akin to those of the tesserae.

All the data produced in this study have been gathered in a QGIS project and available also for the restorers, for a constant monitoring of the areas of intervention and of the related analytical information.

The mosaic is currently undergoing restoration, and new data on the underlying mortars have recently been acquired by the present research group on the backside mortars and are currently being processed. The combination of these new data, along with information obtained from a new instrumental investigation campaign planned for the mosaic surface in the final phases of the restoration operations, will further enrich our knowledge of this superlative work of ancient art.

## Supporting information

**S1 Fig. The distribution of the spectroscopic spot analyses carried out on the Alexander Mosaic.**
(PDF)

**S2 Fig. Representative ER-FTIR (A, C) and Raman (B, D) spectra.**
(PDF)

**S3 Fig. Representative endoscopic images on the back sides of the mosaic.** Evident spaces filled with binder depositions (A), probably composed of vinyl or gypsum-based substances, along with paper, including newspaper (B, C, D).
(PDF)

**S4 Fig. Chromatic variations of mortar observed by means of OM images.**
(PDF)

**S1 Table. Results of Raman (RM) and ER-FTIR (IR) analyses.** Abbreviations: Cal, calcite; Sil, silicates; Gp, gypsum; Ox, oxalates; Brt, baryte; Cer, cerussite.
(XLSX)

**S2 Table. List of pXRF data (net intensities, see text) of the analyzed tesserae.**
(XLSX)

## Acknowledgments

We wish to thank all the MANN staff for the continuous support during this work. This paper greatly benefited from the valuable comments of Prof. Teodosio Donaire and an anonymous Referee. Thanks are also due to the PLOS ONE Associated Editor Prof. Carlos P. Odriozola for helpful suggestions.

## Author Contributions

**Conceptualization:** Antonio De Simone, Alessio Langella, Vincenzo Morra.

**Data curation:** Giuseppina Balassone, Piergiulio Cappelletti, Alberto De Bonis, Diego Di Martire, Sossio Fabio Graziano, Celestino Grifa, Francesco Izzo, Alessio Langella, Mariano Mercurio, Concetta Rispoli, Maria Verde.

**Formal analysis:** Piergiulio Cappelletti, Alberto De Bonis, Diego Di Martire, Sossio Fabio Graziano, Celestino Grifa, Francesco Izzo, Alessio Langella, Mariano Mercurio, Concetta Rispoli, Maria Verde.

**Methodology:** Diego Di Martire, Sossio Fabio Graziano, Celestino Grifa, Francesco Izzo, Alessio Langella, Mariano Mercurio, Concetta Rispoli, Maria Verde.

**Supervision:** Antonio De Simone, Paolo Giulierini, Vincenzo Morra, Mariateresa Operetto, Amanda Piezzo.

**Writing – original draft:** Giuseppina Balassone, Alberto De Bonis, Celestino Grifa, Francesco Izzo, Alessio Langella, Concetta Rispoli, Maria Verde.

**Writing – review & editing:** Giuseppina Balassone, Piergiulio Cappelletti, Alberto De Bonis, Antonio De Simone, Celestino Grifa, Francesco Izzo, Alessio Langella, Concetta Rispoli, Maria Verde.

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
