## [Decision Letter · Decision Letter 0]

30 Aug 2024

PONE-D-24-32594From tiny to immense: geological spotlight on the Alexander Mosaic (National Archaeological Museum of Naples, Italy) using non-invasive in situ analysesPLOS ONE

Dear Dr. Balassone,

Thank you for submitting your manuscript to PLOS ONE. After careful consideration, we feel that it has merit but does not fully meet PLOS ONE’s publication criteria as it currently stands. Therefore, we invite you to submit a revised version of the manuscript that addresses the points raised during the review process.

**Please perform the tasks reviewers ask for and resubmit the paper.**

**I found the paper interesting but lacking the deepness the reviewers are asking for, so please take careful attention to the reviewers and resubmit the paper.**

We look forward to receiving your revised manuscript.

Kind regards,

Carlos P. Odriozola, Ph.D

Academic Editor

PLOS ONE

Journal requirements: 1. When submitting your revision, we need you to address these additional requirements. Please ensure that your manuscript meets PLOS ONE's style requirements, including those for file naming. The PLOS ONE style templates can be found at https://journals.plos.org/plosone/s/file?id=wjVg/PLOSOne_formatting_sample_main_body.pdf and https://journals.plos.org/plosone/s/file?id=ba62/PLOSOne_formatting_sample_title_authors_affiliations.pdf. 2. In your manuscript, please provide additional information regarding the specimens used in your study. Ensure that you have reported human remain specimen numbers and complete repository information, including museum name and geographic location.  If permits were required, please ensure that you have provided details for all permits that were obtained, including the full name of the issuing authority, and add the following statement: 'All necessary permits were obtained for the described study, which complied with all relevant regulations.' If no permits were required, please include the following statement: 'No permits were required for the described study, which complied with all relevant regulations.' For more information on PLOS ONE's requirements for paleontology and archeology research, see https://journals.plos.org/plosone/s/submission-guidelines#loc-paleontology-and-archaeology-research. 3. We note that the grant information you provided in the ‘Funding Information’ and ‘Financial Disclosure’ sections do not match.  When you resubmit, please ensure that you provide the correct grant numbers for the awards you received for your study in the ‘Funding Information’ section. 4. Thank you for stating the following financial disclosure:  [National Archaeological Museum of Naples (MANN), project # 4913-P].  Please state what role the funders took in the study.  If the funders had no role, please state: ""The funders had no role in study design, data collection and analysis, decision to publish, or preparation of the manuscript."" If this statement is not correct you must amend it as needed. Please include this amended Role of Funder statement in your cover letter; we will change the online submission form on your behalf. 5. We note that Figure(s) 1, 2, 3, 4, 6, S1, S2, S4 and S5 in your submission contain copyrighted images. All PLOS content is published under the Creative Commons Attribution License (CC BY 4.0), which means that the manuscript, images, and Supporting Information files will be freely available online, and any third party is permitted to access, download, copy, distribute, and use these materials in any way, even commercially, with proper attribution. For more information, see our copyright guidelines: http://journals.plos.org/plosone/s/licenses-and-copyright. We require you to either (1) present written permission from the copyright holder to publish these figures specifically under the CC BY 4.0 license, or (2) remove the figures from your submission: a. You may seek permission from the original copyright holder of Figure(s) 1, 2, 3, 4, 6, S1, S2, S4 and S5  to publish the content specifically under the CC BY 4.0 license.  We recommend that you contact the original copyright holder with the Content Permission Form (http://journals.plos.org/plosone/s/file?id=7c09/content-permission-form.pdf) and the following text:“I request permission for the open-access journal PLOS ONE to publish XXX under the Creative Commons Attribution License (CCAL) CC BY 4.0 (http://creativecommons.org/licenses/by/4.0/). Please be aware that this license allows unrestricted use and distribution, even commercially, by third parties. Please reply and provide explicit written permission to publish XXX under a CC BY license and complete the attached form.” Please upload the completed Content Permission Form or other proof of granted permissions as an ""Other"" file with your submission.  In the figure caption of the copyrighted figure, please include the following text: “Reprinted from [ref] under a CC BY license, with permission from [name of publisher], original copyright [original copyright year].” b. If you are unable to obtain permission from the original copyright holder to publish these figures under the CC BY 4.0 license or if the copyright holder’s requirements are incompatible with the CC BY 4.0 license, please either i) remove the figure or ii) supply a replacement figure that complies with the CC BY 4.0 license. Please check copyright information on all replacement figures and update the figure caption with source information. If applicable, please specify in the figure caption text when a figure is similar but not identical to the original image and is therefore for illustrative purposes only.

Reviewers' comments:

Reviewer's Responses to Questions

**Comments to the Author**

1. Is the manuscript technically sound, and do the data support the conclusions?

Reviewer #1: Partly

Reviewer #2: Yes

2. Has the statistical analysis been performed appropriately and rigorously? 

Reviewer #1: Yes

Reviewer #2: Yes

3. Have the authors made all data underlying the findings in their manuscript fully available?

Reviewer #1: Yes

Reviewer #2: Yes

4. Is the manuscript presented in an intelligible fashion and written in standard English?

Reviewer #1: Yes

Reviewer #2: Yes

5. Review Comments to the Author

Reviewer #1: The chemical and textural characterization of stone material in artworks is a subject of great importance in restoration processes. There are methodologies, such as petrographic analysis using thin sections, whole rock geochemistry or isotopic analysis, which allow the classification of rocks and the proposal of an area of provenance, but these are destructive methods that would practically require the extraction of numerous pieces. The authors present a complete and exhaustive study of the tesserae of the Alexander Mosaic, but using non-invasive methods. The results obtained constitute an important starting point for determining their state of conservation and for possible restoration processes. Therefore, I consider that this is a relevant study for this journal.

I found that the paper is well written and the authors performed careful and thorough data processing. However, the main objection I have is the description and interpretation of the chemical and textural features of the tesserae. I consider that the different groups are described in a somewhat disordered way, without a basic characterization of the textural features, which does not allow a precise classification and comparison with other lithotypes.

Although the four groups show very heterogeneous textural features, the most common should be indicated: 1) crystallinity (holocrystalline or glassy); 2) relative size of the crystals (equigranular, inequigranular, porphyritic), 3) absolute range of grain size (... medium grain: 1-5 mm; ... fine grain: 1-0.05 mm;... very fine grain: < 0.05 mm; microcrystalline, etc.). For crystalline igneous rocks, the color index can also be used. This index is the percent, by volume, of dark-colored (i.e., mafic: pyroxene, amphibole, biotite, etc.) minerals in a rock. According to this index, rocks may be divided into leucocratic (color index, 0 to 30), mesocratic (color index, 30 to 60), and melanocratic (color index, 60 to 100).

Lines 285-351. Please order the description sections: 1. Indicate which compositional feature differentiates this type from the others, 2. Colors and location in the mosaic, 3. Chemical composition and 4. Textures.

For example:

Lines 318-328. Group 2. silicate-bearing tesserae

This group is formed by tesserae with a silicate-bearing composition displaying grey, black, and green colors (Fig 5). Black tesserae (Fig 4N) are characterized by different sizes. Larger tesserae are part of the outer frame of the mosaic, the smallest one was used to highlight the characters’ outlines and the most distinctive details (e.g., lips, eyebrows and eyes) with the purpose of creating a stronger contrast of the figures.

Most of these tesserae show high intensity of Si (S2 Table), along with Fe, Ca, K, and Ti, likely pyroxene and feldspar bearing rocks, as suggested by vibrational spectroscopy (S1 Table). Some of the green tesserae (Fig 4P), also showing high Si and Fe levels, might point to a possible greenish silicate-rich rocks. They are melanocratic to leucocratic rocks with fine-grained (Fig. 4O) to aphyric/glassy-like textures.

Lines 330-341. Group 3. Please rearrange the description, describe the textures and move the following paragraph to the discussion: “This composition is likely related to vitreous (i.e. synthetic) tesserae, being the green ones richer in Cu than the gray ones (S2 Table). pXRF measurements performed on blue and green vitreous tesserae from other mosaics exposed at MANN (Mosaic Room 59) confirmed this assumption (Fig 5).” (Lines 338-341).

Lines 343-351. Group 4. Please rearrange the description and describe the textures.

Lines 394-457. The section on “geological materials and sourcing of tesserae” should be reorganized to explain the typology and provenance of the four differentiated groups, rather than the provenance by color. Some observations:

Lines 439-441. Marmor Taenarium is an impure marble coloured red by hematite and Figure 6 shows a dark red tessera with intermediate composition and high intensity in Si (Group 3), so this origin should be ruled out. Also in Figure 6, the sample called "black silicate" shows a melanocratic plutonic rock with a microcrystalline equigranular texture similar to gabbroic or dioritic rocks (I would not relate it to a volcanic rock from Vesuvius). Furthermore, the sample "black carbonate" has a porphyritic aphanitic texture, characteristic of a volcanic rock, so it cannot be compared with Nero Antico, which is a black marble. This volcanic texture is also observed in the sample in Figure 4 U. The presence of tesserae with volcanic textures suggests a local provenance.

Minor Issues

Lines 124-132. Rocks are naturally heterogeneous at small scales due to factors such as mineralogical variations or grain size along the rock. Since the tesserae are around 2 mm in size, please indicate the spot size analyzed by the p-XRF and how it can affect the measurements obtained.

Lines 249-252. “These tesserae are joint by a whitish mortar well observed by means of OM images (Fig 4) showing faded chromatic variations resembling those of the tesserae (S5 Fig), likely made to give continuity to the depicted scene as in a painting, as also mentioned in literature [5].”

Figures 4F, 4I and 4J show carbonated tesserae with an evident reaction border in contact with the mortar. Since these tesserae were covered by hot pyroclasts, it is also possible that the coloration of these tesserae and the mortar could be due to reaction processes between both components.

Line 295. “Mg-rich variety of calcite”: dolomite.

Line 429 …. with this rocks (Reference); …. Giallo di Siena stone (Reference).

To conclude, I consider the manuscrit acceptable only after revision and rewriting, particularly after addressing all the petrographic evidence, chemical data, and discussion.

T. Donaire

Reviewer #2: Dear Editor,

The article titled “From tiny to immense: geological spotlight on the Alexander Mosaic (National Archaeological Museum of Naples, Italy) using non-invasive in situ analyses” presents an interesting diagnostic study of a highly significant mosaic. It is certainly of interest to PLOS ONE readers. However, in my opinion, some revisions are necessary:

• The authors should provide more information about the pXRF measurement settings, such as whether a primary filter was used.

• In addition to background subtraction, were the XRF spectra processed, for example, by standardizing with the Compton band? Standardization of this type, or another, is essential when analyzing peak intensities.

• Does the instrumentation used allow for precise observation of the exact point where the analysis is conducted? In the case of very small mosaic tesserae, it is extremely easy to misidentify the analysis point. The authors should also provide information on the spot size for both XRF and Raman measurements.

• Page 9, lines 219-222: It is unclear to what the authors attribute the variations in the infrared thermography images. Are these due to deterioration? If so, what type?

• Which elements were included in the principal component analysis? From the figure, it appears only Ca, Fe, and Pb were considered. If the analysis included more elements, the loading for each should be provided.

• Page 14, line 230: This line should be revised (for example, "Group 3. Tesserae with intermediate composition and vitreous aspect").

• Page 15, line 343: Since iron is present in all the tesserae, the group name should be modified. Furthermore, the authors should clarify the sentence in lines 349-351.

6. PLOS authors have the option to publish the peer review history of their article (what does this mean?). If published, this will include your full peer review and any attached files.

Reviewer #1: No

Reviewer #2: No

---

## [Author Response · Author response to Decision Letter 0]

4 Nov 2024

Rebuttal Letter: manuscript PONE-D-24-32594

Summary of Response to Reviewers and revisions

Associate Editor’s comments:

Please perform the tasks reviewers ask for and resubmit the paper.

I found the paper interesting but lacking the deepness the reviewers are asking for, so please take careful attention to the reviewers and resubmit the paper.

Reply: We thank the Editor for the positive comment on our work. In accordance with his suggestion, we have endeavored to improve the manuscript as indicated by the Reviewers, reporting our point-to-point responses in the next sections.

***

Reviewer #1 

The chemical and textural characterization of stone material in artworks is a subject of great importance in restoration processes. There are methodologies, such as petrographic analysis using thin sections, whole rock geochemistry or isotopic analysis, which allow the classification of rocks and the proposal of an area of provenance, but these are destructive methods that would practically require the extraction of numerous pieces. The authors present a complete and exhaustive study of the tesserae of the Alexander Mosaic, but using non-invasive methods. The results obtained constitute an important starting point for determining their state of conservation and for possible restoration processes. Therefore, I consider that this is a relevant study for this journal.

I found that the paper is well written and the authors performed careful and thorough data processing.

Reply: We are grateful to the Reviewer for positive remarks.

**

However, the main objection I have is the description and interpretation of the chemical and textural features of the tesserae. I consider that the different groups are described in a somewhat disordered way, without a basic characterization of the textural features, which does not allow a precise classification and comparison with other lithotypes.

Although the four groups show very heterogeneous textural features, the most common should be indicated: 1) crystallinity (holocrystalline or glassy); 2) relative size of the crystals (equigranular, inequigranular, porphyritic), 3) absolute range of grain size (... medium grain: 1-5 mm; ... fine grain: 1-0.05 mm;... very fine grain: < 0.05 mm; microcrystalline, etc.). For crystalline igneous rocks, the color index can also be used. This index is the percent, by volume, of dark-colored (i.e., mafic: pyroxene, amphibole, biotite, etc.) minerals in a rock. According to this index, rocks may be divided into leucocratic (color index, 0 to 30), mesocratic (color index, 30 to 60), and melanocratic (color index, 60 to 100).

Reply: We thank the Reviewer for this observation, that contributed to improve our work. We have changed this part, trying to improve the overall clarity of groups description as suggested. However, as we only performed digital videomicroscopy and no sampling for more accurate destructive analyses (i.e., thin section) was allowed, we preferred not to use detailed terms such as melanocratic and leucocratic that are defined based on the proportion of mafic minerals, not easily visible with our non-destructive approach. For the same reason, no grain size can be provided, and only descriptive terms were used. 

**

Lines 285-351. Please order the description sections: 1. Indicate which compositional feature differentiates this type from the others, 2. Colors and location in the mosaic, 3. Chemical composition and 4. Textures.

For example:

Lines 318-328. Group 2. silicate-bearing tesserae

This group is formed by tesserae with a silicate-bearing composition displaying grey, black, and green colors (Fig 5). Black tesserae (Fig 4N) are characterized by different sizes. Larger tesserae are part of the outer frame of the mosaic, the smallest one was used to highlight the characters’ outlines and the most distinctive details (e.g., lips, eyebrows and eyes) with the purpose of creating a stronger contrast of the figures.

Most of these tesserae show high intensity of Si (S2 Table), along with Fe, Ca, K, and Ti, likely pyroxene and feldspar bearing rocks, as suggested by vibrational spectroscopy (S1 Table). Some of the green tesserae (Fig 4P), also showing high Si and Fe levels, might point to a possible greenish silicate-rich rocks. They are melanocratic to leucocratic rocks with fine-grained (Fig. 4O) to aphyric/glassy-like textures.

Reply: We tried to reorganize this section following the Reviewer’s indications as far as possible.

**

Lines 330-341. Group 3. Please rearrange the description, describe the textures and move the following paragraph to the discussion: “This composition is likely related to vitreous (i.e. synthetic) tesserae, being the green ones richer in Cu than the gray ones (S2 Table). pXRF measurements performed on blue and green vitreous tesserae from other mosaics exposed at MANN (Mosaic Room 59) confirmed this assumption (Fig 5).” (Lines 338-341).

Reply: As for the previous comment, we tried to reorganize this section following the Reviewer’s suggestions. A green tessera was substituted in Fig 4T, and the text modified accordingly, to improve the consistency and persuasiveness of this part (please see the new Fig 4 included in this revised version).

**

Lines 343-351. Group 4. Please rearrange the description and describe the textures.

Reply: As for the previous comment, we tried to reorganize this section following the Reviewer’s suggestions.

**

Lines 394-457. The section on “geological materials and sourcing of tesserae” should be reorganized to explain the typology and provenance of the four differentiated groups, rather than the provenance by color. Some observations:

Lines 439-441. Marmor Taenarium is an impure marble coloured red by hematite and Figure 6 shows a dark red tessera with intermediate composition and high intensity in Si (Group 3), so this origin should be ruled out. Also in Figure 6, the sample called "black silicate" shows a melanocratic plutonic rock with a microcrystalline equigranular texture similar to gabbroic or dioritic rocks (I would not relate it to a volcanic rock from Vesuvius). Furthermore, the sample "black carbonate" has a porphyritic aphanitic texture, characteristic of a volcanic rock, so it cannot be compared with Nero Antico, which is a black marble. This volcanic texture is also observed in the sample in Figure 4 U. The presence of tesserae with volcanic textures suggests a local provenance.

Reply: We are grateful to the Reviewer for the insightful comments, so we have extensively revised this section. First, we have reorganized this part based on the four Groups, as suggested by the Reviewer. Hence, we have also modified the Figure 6 accordingly (potential raw materials eliminated due to copyright issue).

Regarding the dark red tessera of Group 3, it has high Si intensity but also the Ca one is remarkable; so, we hypothesize that a provenance of Marmor Taenarium might be still possible. Concerning the black silicate, we have deleted the Vesuvius sourcing, as correctly suggested by the Reviewer; as regards the black carbonate, we have better specified the texture description, so we would like to leave the assignment to the Nero Antico carbonate lithotype as a possible raw material.

Furthermore, we have also tried to improve the green tesserae provenance issue of Group 3.

**

Minor Issues

Lines 124-132. Rocks are naturally heterogeneous at small scales due to factors such as mineralogical variations or grain size along the rock. Since the tesserae are around 2 mm in size, please indicate the spot size analyzed by the p-XRF and how it can affect the measurements obtained.

Reply: We agree with the Reviewer about the importance of the scale of the analyzed points. We have specified the spot size.

**

Lines 249-252. “These tesserae are joint by a whitish mortar well observed by means of OM images (Fig 4) showing faded chromatic variations resembling those of the tesserae (S5 Fig), likely made to give continuity to the depicted scene as in a painting, as also mentioned in literature [5].”

Figures 4F, 4I and 4J show carbonated tesserae with an evident reaction border in contact with the mortar. Since these tesserae were covered by hot pyroclasts, it is also possible that the coloration of these tesserae and the mortar could be due to reaction processes between both components.

Reply: We thank the Reviewer for bringing this issue to our attention. However, we are inclined towards the opinion that the described hues were intentionally made in various parts of the mosaic, to create a chromatic connection between the mortars and the tesserae, as also noted by Mellillo (2013).

**

Line 295. “Mg-rich variety of calcite”: dolomite.

Reply: The information come only from the pXRF analysis that evidenced a higher abundance of Mg. We have no other analytical evidence for discriminating between calcite and dolomite, therefore we preferred to name it, at least in this line, as a generic Mg-carbonate.

**

Line 429 …. with this rocks (Reference); …. Giallo di Siena stone (Reference).

Reply: Done.

**

To conclude, I consider the manuscrit acceptable only after revision and rewriting, particularly after addressing all the petrographic evidence, chemical data, and discussion.

Reply: We tried to extensively revise the whole manuscript according to these Reviewer’s suggestions, particularly about the chemical vs. petrographic description, as well as to the discussion paragraph, to improve the overall organization and clarity of our manuscript.

***

Reviewer #2

The article titled “From tiny to immense: geological spotlight on the Alexander Mosaic (National Archaeological Museum of Naples, Italy) using non-invasive in situ analyses” presents an interesting diagnostic study of a highly significant mosaic. It is certainly of interest to PLOS ONE readers.

Reply: We are grateful to the Reviewer for positive comments and appreciation of our work. Below we report our responses to the specific comments. 

**

However, in my opinion, some revisions are necessary:

• The authors should provide more information about the pXRF measurement settings, such as whether a primary filter was used.

Reply: We thank the Reviewer for pointing this out. Measurements were performed under unfiltered conditions. We have added the requested analytical details.

**

• In addition to background subtraction, were the XRF spectra processed, for example, by standardizing with the Compton band? Standardization of this type, or another, is essential when analyzing peak intensities.

Reply: Corrections including Background and Escape peaks were applied to avoid misidentification of elemental peaks. Compton band was not standardized as it typically looks distinct from elemental peaks.

**

• Does the instrumentation used allow for precise observation of the exact point where the analysis is conducted? In the case of very small mosaic tesserae, it is extremely easy to misidentify the analysis point. The authors should also provide information on the spot size for both XRF and Raman measurements.

Reply: The pXRF instrument is equipped with a camera that allows for precise observation of the analysed point. Spot size is 3 mm for both instruments. Information has been added in the methods. However, as the reviewer rightly points out, some tiles can be very small. In this case, a portion of the jointing mortar outside the tesserae might have been accidentally included in the measurement spot. This aspect was already evidenced in the manuscript (see page 12 and lines #269-270 of the original manuscript): “The occurrence of Ca is mostly ascribed to chemical composition of both carbonate-based components (i.e., tesserae and/or possible contamination of jointing mortars)…”.

**

• Page 9, lines 219-222: It is unclear to what the authors attribute the variations in the infrared thermography images. Are these due to deterioration? If so, what type?

Reply: As suggested by the Reviewer, more details have been added to the thermographic analysis section. 

**

• Which elements were included in the principal component analysis? From the figure, it appears only Ca, Fe, and Pb were considered. If the analysis included more elements, the loading for each should be provided.

Reply: The 16 elements (Al, Ca, Cl, Cr, Cu, Fe, K, Mg, Mn, Ni, Pb, S, Si, Sr, Ti, Zn) were included in the PCA. A zoom view of the loadings area of the other elements has been added to the figure (please see the new Fig 6 included in this revised version).

**

• Page 14, line 230: This line should be revised (for example, "Group 3. Tesserae with intermediate composition and vitreous aspect").

Reply: Thanks for the suggestion, we revised the title as such “Natural and vitreous tesserae with intermediate composition”. In this way, we think to have better clarified the fact that the group includes tesserae of intermediate composition both natural and artificial (vitreous).

**

• Page 15, line 343: Since iron is present in all the tesserae, the group name should be modified. Furthermore, the authors should clarify the sentence in lines 349-351.

Reply: The Reviewer’s observation is right. We renamed the group 4 as “Iron-rich tesserae”.

***

Finally, we made further corrections, which are listed below:

- minor stylistic issues have been modified and marked in the manuscript directly;

- minor typos have been corrected and marked in the manuscript directly;

- a very small change was made in Table 1 (e.g., the four Groups better separated);

- in Figure 6, the images of the lower rows (potential raw materials) for provenance issues were eliminated (due to copyright issues) and we maintained the lithological types only, as explained in the text;

- two new literature citations have been added and the reference list has been updated and checked.

---

## [Decision Letter · Decision Letter 1]

22 Nov 2024

From tiny to immense: geological spotlight on the Alexander Mosaic (National Archaeological Museum of Naples, Italy) using non-invasive in situ analyses

PONE-D-24-32594R1

Dear Dr. Balassone,

We’re pleased to inform you that your manuscript has been judged scientifically suitable for publication and will be formally accepted for publication once it meets all outstanding technical requirements.

Kind regards,

Carlos P. Odriozola, Ph.D

Academic Editor

PLOS ONE

Reviewers' comments:

Reviewer's Responses to Questions

**Comments to the Author**

1. If the authors have adequately addressed your comments raised in a previous round of review and you feel that this manuscript is now acceptable for publication, you may indicate that here to bypass the “Comments to the Author” section, enter your conflict of interest statement in the “Confidential to Editor” section, and submit your "Accept" recommendation.

Reviewer #1: All comments have been addressed

Reviewer #2: All comments have been addressed

2. Is the manuscript technically sound, and do the data support the conclusions?

Reviewer #1: Yes

Reviewer #2: Yes

3. Has the statistical analysis been performed appropriately and rigorously? 

Reviewer #1: Yes

Reviewer #2: Yes

4. Have the authors made all data underlying the findings in their manuscript fully available?

Reviewer #1: Yes

Reviewer #2: Yes

5. Is the manuscript presented in an intelligible fashion and written in standard English?

Reviewer #1: Yes

Reviewer #2: Yes

6. Review Comments to the Author

Reviewer #1: (No Response)

Reviewer #2: the authors have responded to the comments in an exhaustive manner, improving, in my opinion, the quality of the work that can be accepted for publication

7. PLOS authors have the option to publish the peer review history of their article (what does this mean?). If published, this will include your full peer review and any attached files.

Reviewer #1: **Yes: **Teodosio Donaire

Reviewer #2: No

---

## [Editor Report · Acceptance letter]

29 Nov 2024

PONE-D-24-32594R1 

PLOS ONE

Dear Dr. Balassone, 

I'm pleased to inform you that your manuscript has been deemed suitable for publication in PLOS ONE. Congratulations! Your manuscript is now being handed over to our production team.

Kind regards, 

on behalf of

Dr. Carlos P. Odriozola 

Academic Editor

PLOS ONE